# Economic Development and Gender Ratio Change in Chinese Suicide Rates (1990–2017)

**DOI:** 10.3390/ijerph192315606

**Published:** 2022-11-24

**Authors:** Jie Zhang, Juncheng Lyu, Dorian A. Lamis

**Affiliations:** 1School of Public Health, Shandong University, Jinan 250061, China; 2Department of Sociology, State University of New York at Buffalo State, Buffalo, NY 14222, USA; 3School of Public Health, Weifang Medical University, Weifang 261053, China; 4Department of Psychiatry and Behavioral Sciences, Emory University School of Medicine, Atlanta, GA 30307, USA

**Keywords:** gender ratio, suicide, China, strains, economy

## Abstract

*Objective*: The overall gender ratio in Chinese suicide rates has substantially changed during the past three decades. In this study, we investigated the social economic factors and the mechanisms that may be contributing to this fluctuation. *Study Design*: This is a secondary analysis using suicide mortality data from the China Centers for Disease Control and Prevention. *Methods*: A statistical model was performed with province as the unit of analysis. The per capita GDP and income of each provincial-level region were collected from the Economic and Statistical Yearbook. Rate and ratio were used to describe the trend of variations, and correlation analyses were conducted to examine the association between economic development and gender ratio change. *Results*: The China overall male to female gender ratio of suicide rates increased as the GDP per capita grew (r = 0.439; *p* = 0.015). The gender ratio changed from 0.88 in 1990 to 1.56 in 2017, with the reversion point between 1995 and 2000. The most radical reverse changes in the gender ratios were found in large municipalities. *Conclusions*: Cultural and social economic variables may explain the gender ratio changes. Increased economic development has significantly reduced psychological strains on rural young women, which in turn decreased the suicide rate among this sub-population.

## 1. Introduction

The overall suicide rate in China was approximately 23/100,000 in the late 1990s and was one of the highest in the world [1]. Moreover, China was the only country in the world in which female suicide rates were higher than those of males, with a gender ratio of 0.88 [2]. However, in the past two decades, the Chinese suicide rates have steadily and significantly decreased, with an overall rate of about 7.21/100,000 in 2017. Additionally, the gender ratio has also reversed from 0.88 to 1.56, with suicide rates for males becoming higher than those for females [3]. In this study, we identified social economic development factors that may have contributed to these changes.

The average suicide rate in the world remains about 11/100,000, with males dying by suicide approximately 2–4 times more than females [4]. In the United States, the overall suicide rates increased from about 10.5/100,000 in 1990 to about 14/100,000 in 2017, with no change in the gender ratio [5]. Thus, it is important to understand why the suicide rates in China have dropped so rapidly and how the gender ratio has dramatically changed.

Previous studies have addressed social economic factors that are related to the rapid drop of anomalous suicide rates in China [2,3]. Different from the Durkheimian theorem that economic growth and modernity increase suicide rates [6], the findings in Chinese society demonstrated that fast economic development with improved living conditions and increased opportunities reduces individual psychological strains, which decreased suicidality.

Previous studies also have indicated that some contributing factors were associated with the gender ratio of suicide. Chang Q indicated that high sex ratio at birth was significantly associated with a lower suicide gender ratio [7]. Prior literature has indicated that the gender ratio of suicide in China has increased over the past three decades [3]. Jiang reported that male suicide rates were increasingly higher than females’ after 2006 in China [8]. However, many previous reports did not interpret the potential reasons for these changes based on social theory.

The strain theory of suicide is often used to account for the increase or decrease in suicide rates [2]. Specifically, the theory posits that a suicidal behavior is likely to be preceded by psychological strains (frustration, hopelessness, desperation) resulting from negative life events in social structure and the environment. A strain is at its minimum a frustration beyond a single or simple stress. A strain consists of two or more conflicting stresses pulling the individual in different directions. There are four sources of psychological strains: (1) differential social values; (2) discrepancy between aspiration and reality; (3) relative deprivation; and (4) deficient coping skills or resources [9]. The strain theory of suicide may also explain the change in gender ratio of suicide in China.

In this study, we focus on the change of gender ratios in Chinese suicide rates during the past three decades, with the hypothesis that the higher the GDP per capita, the lower the psychological strains and the higher the suicide gender ratio. The mechanism behind the correlation is assumed to be the reduction of psychological strains, particularly among Chinese rural young women.

## 2. Materials and Methods

Under the auspices of the China National Health Commission, the database of causes of death was generally composed of vital registration (VR), verbal autopsy (VA), registry, survey, police, and surveillance data. The Global Burden of Diseases (GBD), Injuries, and Risk Factors Study protocols were used to standardize the causes of mortality data [10,11].

Using the data of suicide mortality first released by the Chinese Centers for Disease Control and Prevention (China CDCs) from 1990 to 2017, we examined the suicide rate changes for each of the 33 provinces. Taiwan health and mortality data were not reported to the Chinese CDC and therefore are unavailable for the current study. Data of suicide in each area were aggregated every five years, except for the year of 2017, and the 5-year age groups from 5–9 to 90–94 years and ≥95 years were recorded in the database. More information about the mortality data and the methods in collecting the data can be found in past publications [3,11].

The economic growth data such as GDP per capita and income per capita for each of the study years and for each of the provincial level regions were also obtained from the China Statistical Yearbook [12].

The means were used to describe the variations in suicide rates. The rate and ratio were used to investigate the trend of variations. We calculated the Pearson correlation coefficients to explore and estimate the effects of each of the economic factors on the suicide rates and changes.

## 3. Results

Based on the China Statistical Yearbook, the GDP per capital and income per capita for each of the year periods from 1990 to 2017 and for each of the provincial regions in China steadily increased.

The suicide rates by gender for each of the year periods and for each of the provincial level regions are listed in Table 1. The table also illustrates the changes in the rates from 1990 to 2017. In comparison of gender, the majority of the provincial level regions had a more decreased change in suicide rates for females than for males. Exceptions were found for the provinces Gansu, Hebei, Liaoning, and Shaanxi, where more decreased changes in suicide rates were found in males as compared to females, due to the more rural and less developed nature of these regions. Furthermore, Hong Kong and Macao did not fit this pattern, as they had higher suicide rates for males than for females at the first study year.

In the calculations, the distribution probability measure (*p*) was used together with the Pearson contingency coefficient (C). The point-biserial correlation coefficient was also determined, denoted as r (Pearson’s correlation coefficient), for the relation among GDP per capita increase, income per capita increase, and the suicide rate gender ratio increase, and the correlation matrix can be found in Table 2. Among the Chinese provincial-level regions, the suicide rate gender ratio increase is associated with the increase in the GDP per capita (r = 0.439, *p* = 0.015), and there is a moderate relationship between these variables. A trend for the gender ratio to increase along with the increased income per capita was found, and there is a weak relationship between them, although it was not statistically significant (r = 0.282, *p* = 0.124).

Changes in the gender ratios of suicide rates from 1990 to 2017 were calculated by subtracting the gender ratio of 1990 from the gender ratio of 2017, and Table 3 shows the numbers for each of the provincial-level regions. The greatest increase in the gender ratio was found in Beijing, Shanghai, and Tianjin, the three municipality cities.

In summary, the suicide rates decreased, the GDP per capita increased, the income per capita increased, and the suicide gender ratio increased for each province from 1990 to 2017. Detailed information leading to the findings will be available upon request to the authors.

## 4. Discussion

To investigate the factors behind the inversed change in the suicide rate gender ratios, we analyzed the relationship between gender ratios of suicide and GDP per capita. We found that with rapid development of economic growth, increased GDP per capita was an important correlate factor (r = 0.439; *p* = 0.015) for gender ratio of suicide rate. From 1990 to 2017, the suicide rates in China have steadily and significantly dropped in all of the 33 provincial level regions. As the drop in female suicide rates was greater than in male suicide rates in almost all regions, the overall suicide rate gender ratio has significantly increased over the past three decades. Previous studies indicated that rural dwelling is known to be related to higher suicide rates [13,14]. But with regards to economic development, suicide rates in the world usually go up with economic growth including modernization, urbanization, and industrialization [6]. Contrary to the common understanding, suicide rates in China did not fall into this familiar pattern. The strain theory of suicide explains not only the rise in the suicide rates in China three decades ago, but also accounts for the drop in the overall rates in China today.

Based on the theoretical frameworks established by previous sociologists on deviance such as Merton [15] and Agnew [16], the strain theory of suicide examines suicide as a deviant behavior and reveals how a suicidal mindset develops following a negative life event [17]. A psychological strain consists of at least two forces, or two stresses, which push or pull the same person to different directions. When a person experiences two or more differential social values, the value strain is likely to come into being. When someone’s life goal or current expectation is much higher than what the person has, aspiration strain happens. When people living in the same social setting have access to different resources, the relatively less privileged experience deprivation strain. When people encounter a crisis, those who are lacking coping skills and/or resources are more likely to experience the crisis strain [18]. The self-perceived strain makes the individual so uncomfortable, frustrated, hopeless, helpless, or even angered that something has to be done to release the tension. The outward release of the frustration may lead to violence toward others such as crime, and the inward vent of the tension may cause one to self-harm and/or attempt suicide [19].

The strain theory of suicide explains the extraordinarily high suicide rates in China with value strain, aspiration strain, and relative deprivation, as well as the crisis strain among Chinese people three decades ago. It also may explain why the female suicide rate, particularly among rural young females, was much higher than in other youth populations [2].

It is well known that economic level often has an effect on social surroundings, spiritual perception, and people‘s concepts. The fast economic development in China accounts for the rapid decline in Chinese suicide rates [3], and it may also explain the inversed and increased gender ratio in Chinese suicide rates. When the Chinese suicide rates were among the highest in the world about three decades ago, the suicide numbers of rural young women were a major contributor. With more suicides being female than male, the gender ratio was then 0.88. Now, the ratio has increased to 1.56 in 2017, with the reversion point between 1995 and 2000. The number of female suicides was reduced at a higher magnitude than that of the male suicides.

Since China opened its door to the West in 1980s, traditional values such as Confucian teachings have been declining among youths as China has become more globalized and modernized. The traditional values concerning marriage, dating, filial piety, and women’s status that were in conflict with modern Western ideas are weakening among young people [18]. Therefore, the psychological strains formed by conflicting values have become less prevalent in this high-risk and rural young female population. Generally, there was a long lag or buffer period for social development and the change of people’s concepts, so the reversion point of gender ratio was revealed to be between 1995 and 2000. There are now additional opportunities for rural women, who now have more freedom to move to cities and obtain an education and employment [2].

There have been many studies on deprivation and suicide behavior. For example, the extant research has indicated that psychosocial stress was a potential mechanism underlying the link between relative deprivation and suicide mortality in South Korea [20], socioeconomic deprivation was associated with suicide mortality in a Dutch register-based case-control study [21], suicide rates were significantly higher in municipalities with higher levels of deprivation in Japan [22], and psychological strains increased the odds of suicide death in China [23]. In all of these studies, the reduction of psychological strains decreased suicidality.

Our previous research has confirmed that the high economic development rapidly decreased psychological strains in China, especially among females [17]. Economic prosperity and improved living conditions have benefited the Chinese people, especially Chinese women. Rapidly improved quality of life has reduced people’s strains (conflicting stressors) that used to be from deprivation and frustrated aspirations. Based on the strain theory of suicide, the lack of psychological strains, frustration, anger, and psychological pain all contributed to a decrease in suicide risk [24].

The strain theory of suicide explains not only the high suicide rates in China three decades ago, but also the decline in the rates and the change in gender ratio of the suicide rate today. Lack of development opportunities, disadvantaged living conditions, and confused social values increased psychological strains and suicide risk for the Chinese people in the 1990s. Today in China, available opportunities, improved life standards, economic prosperity, and decreased value conflicts have successfully reduced psychological strains in China, particularly for Chinese women. The fast economic prosperity has reduced the Chinese suicide rates, particularly for Chinese women, which has resulted in the significant change in the suicide gender ratio.

## 5. Limitations

Data on suicide is based on a large-sampled surveillance system, which may be limited by quality flaws, which may lead to bias. The current study is an ecological study, and thus, the ecological bias is inevitable. Another economic factors or variables such as inflation, indicators of sustainable development, etc., should be included in the further study.

## 6. Conclusions

Cultural and social economic variables may explain the gender ratio changes. Increased economic development has significantly reduced psychological strains on rural young women, which in turn decreased the suicide rate among this sub-population. Socio-economic factors (such as wealth, economic level, cognitive concept, etc.) are one of the most effective methods of limiting suicidal behavior. Therefore, the local government and community doctors should screen out the main economic and cultural causes of psychological strains and try their best to reduce them so as to promote the decrease of suicide behaviors in the region and community.

## Figures and Tables

**Table 1 ijerph-19-15606-t001:** Suicide rates of each province by gender and the decreases from 1990 to 2017.

No.	Province	Gender	Years							Mean	Suicide Rate Decreased from 1990 to 2017
			1990	1995	2000	2005	2010	2015	2017		
1	Anhui	Male	31.99	31.63	25.85	19.51	14.26	12.12	12.13	21.07	19.86
		Female	44.82	41.92	25.40	16.08	12.72	9.38	8.9	22.74	35.92
2	Beijing	Male	8.09	7.86	7.11	4.76	3.98	3.84	3.9	5.65	4.19
		Female	6.68	6.23	3.94	2.33	1.59	1.22	1.19	3.31	5.49
3	Chongqing	Male	18.19	15.63	10.83	8.31	7.19	7.09	7.21	10.64	10.98
		Female	22.98	18.96	10.48	6.76	5.36	4.53	4.43	10.5	18.55
4	Fujian	Male	14.65	15.37	14.37	11.64	9.16	7.39	7.26	11.41	7.39
		Female	35.52	30.61	16.29	9.98	8.07	5.80	5.30	15.94	30.22
5	Gansu	Male	34.28	28.63	18.88	13.65	11.81	9.96	9.92	18.16	24.36
		Female	25.87	25.69	16.86	13.36	10.25	6.72	6.3	15.01	19.57
6	Guangdong	Male	13.29	12.77	10.07	7.84	6.37	5.27	5.18	8.68	8.11
		Female	11.49	12.09	7.85	6.1	4.45	2.88	2.62	6.78	8.87
7	Guangxi	Male	14.57	14.78	11.05	8.51	7.59	6.77	6.87	10.02	7.70
		Female	15.32	14.64	7.89	5.45	4.66	3.85	3.73	7.93	11.59
8	Guizhou	Male	22.93	22.59	16.97	14.05	11.95	10.92	10.95	15.77	11.98
		Female	25.44	23.84	14.13	10.45	8.68	6.56	6.21	13.62	19.23
9	Hainan	Male	18.33	16.73	13.80	10.76	9.25	8.57	8.7	12.31	9.63
		Female	20.36	18.88	13.36	8.97	7.32	6	5.89	11.54	14.47
10	Hebei	Male	15.14	14.61	11.98	9.63	8.97	9.08	9.39	11.26	5.75
		Female	8.07	8.29	7.77	6.48	5.97	5.28	5.31	6.74	2.76
11	Heilongjiang	Male	12.76	12.6	11.81	9.76	7.41	6.86	6.9	9.73	5.86
		Female	11.23	10.19	7.71	6.22	4.17	3.25	3.03	6.54	8.20
12	Henan	Male	20.01	20.14	15.88	13.35	11.93	10.56	10.43	14.61	9.58
		Female	19.63	19.14	12.70	10.38	9.37	7.49	6.94	12.24	12.69
13	Hubei	Male	37.87	37.94	29.62	29.1	24.97	20.31	19.84	28.52	18.03
		Female	43.12	45.64	34.11	29.45	25.43	19.49	17.94	30.74	25.18
14	Hunan	Male	17.99	18.12	16.05	15.2	12.36	11.45	11.48	14.66	6.51
		Female	39.29	33.33	16.47	11.46	9.4	7.37	7.03	17.76	32.26
15	Inner Mongolia	Male	21.64	20.63	15.76	12.42	10.13	8.25	8.06	13.84	13.58
		Female	22.24	20.4	11.73	7.65	5.49	4.09	3.83	10.78	18.41
16	Jiangsu	Male	15.16	14.46	12.11	9.5	7.53	6.56	6.46	10.26	8.70
		Female	14.71	15.05	10.48	7.83	5.3	4.47	4.15	8.86	10.56
17	Jiangxi	Male	23.9	21.99	15.13	11.34	9.73	8.41	8.21	14.1	15.69
		Female	36.61	30.32	14.27	8.95	7.18	5.25	4.9	15.35	31.71
18	Jilin	Male	13.57	13.91	11.75	9.39	7.21	6.87	6.94	9.95	6.63
		Female	10.01	9.46	7.14	5.6	4.2	3.15	3.01	6.08	7.00
19	Liaoning	Male	12.73	13.89	11.52	9.93	7.34	7.06	7.03	9.93	5.70
		Female	8.71	8.22	6.88	5.75	4.07	3.73	3.52	5.84	5.19
20	Ningxia	Male	13.15	12.62	9.77	8.12	6.98	6.37	6.35	9.05	6.80
		Female	18.4	16.44	9.75	6.82	5.41	4.02	3.87	9.24	14.53
21	Qinghai	Male	22.88	23.27	18.21	13.41	11.12	10.04	9.98	15.56	12.90
		Female	23.4	24.03	15.71	10.54	8.55	6.49	6.16	13.55	17.24
22	Shaanxi	Male	11.48	11.76	10.71	9.63	7.93	7.48	7.56	9.51	3.92
		Female	8.5	8.94	7.84	6.84	6.24	5.14	4.99	6.93	3.51
23	Shandong	Male	33.59	31.98	20.75	15.54	11.48	9.75	9.55	18.95	24.04
		Female	35.4	33.94	18.65	13	9.67	7.81	7.33	17.97	28.07
24	Shanghai	Male	7.17	7.36	6.52	5.08	4.57	4.61	4.68	5.71	2.49
		Female	6.46	6.28	3.86	2.33	1.96	1.58	1.55	3.43	4.91
25	Shanxi	Male	14.5	13.2	9.42	6.93	6.3	6.33	6.37	9.01	8.13
		Female	12.36	10.76	6.16	3.98	3.46	2.88	2.78	6.06	9.58
26	Sichuan	Male	16.36	17.11	14.77	12.91	9.87	8.82	8.87	12.67	7.49
		Female	16.53	17.03	12.68	10.19	7.69	6.09	5.91	10.88	10.62
27	Tianjin	Male	11.78	10.89	8.23	6.12	5.73	5.86	5.84	7.78	5.94
		Female	9.84	8.16	4.78	3.33	2.76	2.26	2.2	4.76	7.64
28	Tibet	Male	21.68	22.42	17.17	10.91	8.87	7.75	7.9	13.81	13.78
		Female	18.5	18.69	12.00	7.81	5.55	4.29	4.12	10.14	14.38
29	Xinjiang	Male	11.33	11.73	11.02	8.67	7.5	7.35	7.55	9.31	3.78
		Female	11.11	10.65	8.39	6.17	5.36	4.89	4.77	7.33	6.34
30	Yunnan	Male	34.57	35.82	26.80	20.65	17.89	15.58	15.27	23.8	19.30
		Female	40.76	41.69	24.77	17.07	13.66	10.27	9.58	22.54	31.18
31	Zhejiang	Male	13.14	12.92	12.37	10.72	7.73	5.94	5.8	9.8	7.34
		Female	25.18	21.69	10.98	7.69	5.37	3.57	3.29	11.11	21.89
32	Hong Kong SAR	Male	12.76	13.3	13.48	12.74	9.52	8.55	8.51	11.26	4.25
		Female	9.22	8.9	7.90	7.66	6.05	5.35	5.34	7.2	3.88
33	Macao SAR	Male	15.82	14.3	11.59	9.27	8.18	7.38	7.4	10.56	8.42
		Female	12.41	10.15	8.48	7.48	7.22	6.07	5.91	8.24	6.50
All	China	Male	19.65	19.41	15.26	12.47	10.09	8.88	8.82	13.51	10.83
		Female	22.45	21.49	13.37	9.81	7.75	6.01	5.65	12.36	16.80

**Table 2 ijerph-19-15606-t002:** Correlations for GDP per capita increased, income per capita increased, and the suicide rate gender ratio among the 33 provincial-level regions from 1990 to 2017.

Correlation Variables	GDP per Capita Increased	Income per Capita Increased	Suicide Rates Gender Ratio Increased
GDP per capita increased		r = 0.728, *p* < 0.001	r = 0.439, *p* = 0.015
Income per capital increased			r = 0.282, *p* = 0.124

**Table 3 ijerph-19-15606-t003:** Changes in gender ratio of suicide rates for each province from 1990 to 2017.

No.	Province	Gender Ratio							Change of Gender Ratio (2017 − 1990)
		1990	1995	2000	2005	2010	2015	2017	
1	Anhui	0.71	0.75	1.02	1.21	1.12	1.29	1.36	0.65
2	Beijing	1.21	1.26	1.80	2.04	2.50	3.15	3.28	2.07
3	Chongqing	0.79	0.82	1.03	1.23	1.34	1.57	1.63	0.84
4	Fujian	0.41	0.50	0.88	1.17	1.14	1.27	1.37	0.96
5	Gansu	1.33	1.11	1.12	1.02	1.15	1.48	1.57	0.24
6	Guangdong	1.16	1.06	1.28	1.29	1.43	1.83	1.98	0.82
7	Guangxi	0.95	1.01	1.40	1.56	1.63	1.76	1.84	0.89
8	Guizhou	0.9	0.95	1.20	1.34	1.38	1.66	1.76	0.86
9	Hainan	0.9	0.89	1.03	1.20	1.26	1.43	1.48	0.58
10	Hebei	1.88	1.76	1.54	1.49	1.50	1.72	1.77	−0.11
11	Heilongjiang	1.14	1.24	1.53	1.57	1.78	2.11	2.28	1.14
12	Henan	1.02	1.05	1.25	1.29	1.27	1.41	1.5	0.48
13	Hubei	0.88	0.83	0.87	0.99	0.98	1.04	1.11	0.23
14	Hunan	0.46	0.54	0.97	1.33	1.31	1.55	1.63	1.17
15	Inner Mongolia	0.97	1.01	1.34	1.62	1.85	2.02	2.1	1.13
16	Jiangsu	1.03	0.96	1.16	1.21	1.42	1.47	1.56	0.53
17	Jiangxi	0.65	0.73	1.06	1.27	1.36	1.60	1.68	1.03
18	Jilin	1.36	1.47	1.65	1.68	1.72	2.18	2.31	0.95
19	Liaoning	1.46	1.69	1.67	1.73	1.80	1.89	2	0.54
20	Ningxia	0.71	0.77	1.00	1.19	1.29	1.58	1.64	0.93
21	Qinghai	0.98	0.97	1.16	1.27	1.30	1.55	1.62	0.64
22	Shaanxi	1.35	1.32	1.37	1.41	1.27	1.46	1.52	0.17
23	Shandong	0.95	0.94	1.11	1.20	1.19	1.25	1.3	0.35
24	Shanghai	1.11	1.17	1.69	2.18	2.33	2.92	3.02	1.91
25	Shanxi	1.17	1.23	1.53	1.74	1.82	2.20	2.29	1.12
26	Sichuan	0.99	1.00	1.16	1.27	1.28	1.45	1.5	0.51
27	Tianjin	1.2	1.33	1.72	1.84	2.08	2.59	2.65	1.45
28	Tibet	1.17	1.20	1.43	1.40	1.60	1.81	1.92	0.75
29	Xinjiang	1.02	1.10	1.31	1.41	1.40	1.50	1.58	0.56
30	Yunnan	0.85	0.86	1.08	1.21	1.31	1.52	1.59	0.74
31	Zhejiang	0.52	0.60	1.13	1.39	1.44	1.66	1.76	1.24
32	Hong Kong SAR	1.38	1.49	1.71	1.66	1.57	1.60	1.59	0.21
33	Macao SAR	1.27	1.41	1.37	1.24	1.13	1.22	1.25	−0.02
All	China	0.88	0.90	1.14	1.27	1.30	1.48	1.56	0.68

## Data Availability

The data would be available based on reasonable requirement.

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
