# Peer review of "Economic Development and Gender Ratio Change in Chinese Suicide Rates (1990–2017)"

_ijerph, 2022, doi:10.3390/ijerph192315606_

Round 1
Reviewer 1 Report
Since there is a lot of reference to economic growth, perhaps it would have been indicated to present, in some data, perhaps indicators of sustainable development, the economic level of each region.
Secondly, in the conclusions it should be shown who uses this study (very interesting by the way), for what purpose it was carried out, etc.
Author Response
Firstly, thank you very much for your review and I appreciate your contribution and advices for the manuscript.
Comments and Suggestions for Authors
1.Since there is a lot of reference to economic growth, perhaps it would have been indicated to present, in some data, perhaps indicators of sustainable development, the economic level of each region.
Responses: We agree to the viewpoints of the reviewer. There are many indicators to descript the economic growth. But we only used the main index (such as GDP per capita increased and income per capita increased) in the current study. In the future study, we would add those indicators mentioned by the reviewer. I have added this point in the limitation.
Limitations: Data on suicide is based on a large-sampled surveillance system, which may be limited by quality flaws, which may lead to bias. The current study is an ecological study, and thus, the ecological bias is inevitable. Another economic factors or variables such as inflation, indicators of sustainable development etc. should be included in the further study.
2.Secondly, in the conclusions it should be shown who uses this study (very interesting by the way), for what purpose it was carried out, etc.
Responses: Yes, thanks for the reviewer `s good advice. It is very benefit to the application of this study. And I have added some sentence for the applications and the purpose of this study.
The sentence added is as following: Therefore, the local government and community doctors should screen out the main economic and cultural causes of psychological strains, and try their best to reduce these psychological strains, so as to promote the decrease of suicide behaviors in the region and community.

Reviewer 2 Report
It is a fairly well-written scientific article devoted to the subject of suicides, based on the Chinese experience. The research method of Desk Research was used in the scientific analysis, and its use in this analysis is justified, allowing for quite detailed research.
As the results of the presented analyzes show, there is a relationship between the economic development of China in the years 1990-2017 and the decrease in the suicide rate in the female population. And this is important information from the suicidological point of view, clearly showing that the improvement of the socio-economic situation of the population reduces the scale of the phenomenon of suicidal behavior. One can only hope that the conclusions contained in the article are objective, and not limited to just one society, which is Chinese society.
But I also see a few shortcomings that need to be corrected in order for the article to be typically scientific. These are:
1. Paragraph 41-45. It should be added that the analysis of suicidal behavior concerns only one type of suicide, which is anomalous suicide (because the presented analyzes do not concern all types of suicides, including the selfish, altruistic and fatalistic ones, as Emile Durkheim mentioned), as the conclusion of the analyzes leads to.
2. Paragraph 101-107. It should be added that the analysis used the χ2 measure, and in the calculations the (test) distribution probability measure (p) was used together with the Pearson contingency coefficient (C). The point-biserial correlation coefficient was also determined, denoted as r (Pearson's correlation coefficient). It should be indicated, when analyzing the values shown (r = 0.439, p = 0.015 - paragraph 105), that there is a weak relationship between these variables. Obviously, this does not diminish the value of the quoted conclusions from the analyzes, but also does not allow us to consider that it is a significant conclusion.
3. Paragraph 96. The text states that Shanxi shows a change in the gender suicide tendency compared to Shaanxi in Tables 1 and 3. This should be checked and corrected according to the facts for a better analysis of the issue.
4. Paragraph 203. In conclusions it can be added that socio-economic factors (wealth) are one of the most effective methods of limiting suicidal behavior.
Author Response
Firstly, thank you very much for your review and I appreciate your contribution and advices for the manuscript.
Comments and Suggestions for Authors
It is a fairly well-written scientific article devoted to the subject of suicides, based on the Chinese experience. The research method of Desk Research was used in the scientific analysis, and its use in this analysis is justified, allowing for quite detailed research.
Responses: I appreciate your recognition and commendation.
As the results of the presented analyzes show, there is a relationship between the economic development of China in the years 1990-2017 and the decrease in the suicide rate in the female population. And this is important information from the suicidological point of view, clearly showing that the improvement of the socio-economic situation of the population reduces the scale of the phenomenon of suicidal behavior. One can only hope that the conclusions contained in the article are objective, and not limited to just one society, which is Chinese society.
But I also see a few shortcomings that need to be corrected in order for the article to be typically scientific. These are:
1.Paragraph 41-45. It should be added that the analysis of suicidal behavior concerns only one type of suicide, which is anomalous suicide (because the presented analyzes do not concern all types of suicides, including the selfish, altruistic and fatalistic ones, as Emile Durkheim mentioned), as the conclusion of the analyzes leads to.
Responses: Thanks to the expert's professional explanation, we have added the description according to the expert advice.
2.Paragraph 101-107. It should be added that the analysis used the χ2 measure, and in the calculations the (test) distribution probability measure (p) was used together with the Pearson contingency coefficient (C). The point-biserial correlation coefficient was also determined, denoted as r (Pearson's correlation coefficient). It should be indicated, when analyzing the values shown (r = 0.439, p = 0.015 - paragraph 105), that there is a weak relationship between these variables. Obviously, this does not diminish the value of the quoted conclusions from the analyzes, but also does not allow us to consider that it is a significant conclusion.
Responses: Yes, I agree to add the description of statistical methods and added the exposition of the degree /level of relationship. The revised version is as following:
In the calculations the distribution probability measure (p) was used together with the Pearson contingency coefficient (C). The point-biserial correlation coefficient was also determined, denoted as r (Pearson's correlation coefficient), for the relation among GDP per capita increase, income per capita increase, and the sui-cide rate gender ratio increase, and the correlation matrix can be found in Table 2. Among the Chinese provincial level regions, the suicide rate gender ratio increase is associated with the increase of the GDP per capita (r=0.439, p=0.015), there is is a moderate relationship between these variables. A trend for the gender ratio to increase along with the increased income per capita was found, there is a weak relationship between them, although it was not statistically significant (r=0.282, p=0.124).
3.Paragraph 96. The text states that Shanxi shows a change in the gender suicide tendency compared to Shaanxi in Tables 1 and 3. This should be checked and corrected according to the facts for a better analysis of the issue.
Responses: Thanks you for the check of data carefully. I have checked according to the tables and have verified Shanxi to Shaanxi. Thanks again.
- Paragraph 203. In conclusions it can be added that socio-economic factors (wealth) are one of the most effective methods of limiting suicidal behavior.
Responses: Thanks for your good advice. The raw conclusion is so sample, your suggestion is fully welcomed to be added in the conclusions part. We have added it in the conclusions part. The revised version is as following:
Conclusions: Cultural and social economic variables may explain the gender ratio changes. In-creased economic development has significantly reduced psychological strains on rural young women, which in turn decreased the suicide rate among this sub-population. So-cio-economic factors (such as wealth, economic level, cognitive concept etc.) are one of the most effective methods of limiting suicidal behavior. Therefore, the local government and community doctors should screen out the main economic and cultural causes of psychological strains, and try their best to reduce them, so as to promote the decrease of suicide behaviors in the region and community.

Reviewer 3 Report
The paper itself elaborates on an exciting topic, but it should be better elaborated. If we correlate the suicide rate and GDP in almost 30 years, we need to incorporate inflation at least.
The literature review is poor, and it is more than evident that women become empowered in more modern society, so the decrease in the female suicide rate is not surprising, or it is not elaborated in depth. Also, some information related to the survey is available upon request—nothing new or of scientific interest to the international public for sure.
Author Response
Firstly, thank you very much for your review and I appreciate your contribution and advices for the manuscript.
1.The paper itself elaborates on an exciting topic, but it should be better elaborated. If we correlate the suicide rate and GDP in almost 30 years, we need to incorporate inflation at least.
Responses: Yes, you are right; the inflation indeed would be a result in bias in the research. Firstly, this is a study in a same country and the difference of inflation among different province is not so large in China. Secondly, we have added this point in the limitation and shortcoming part.
Limitations: Data on suicide is based on a large-sampled surveillance system, which may be limited by quality flaws, which may lead to bias. The current study is an ecological study, and thus, the ecological bias is inevitable. Another economic factors or variables such as inflation, indicators of sustainable development etc. should be included in the further study.
2.The literature review is poor, and it is more than evident that women become empowered in more modern society, so the decrease in the female suicide rate is not surprising, or it is not elaborated in depth. Also, some information related to the survey is available upon request—nothing new or of scientific interest to the international public for sure.
Responses: We agree to the viewpoints of the reviewer. There are many aspect affect the suicide rate. This current study only pays attention to the economic level (only used part main economic index) on suicide. This is a second data analysis and data came from yearbooks so there is not included another variables related to the survey. So it may be another limitation of the current study, based on the suggestion of reviewer, we have added this point in the limitation. Thanks for your understanding.
Limitations: Data on suicide is based on a large-sampled surveillance system, which may be limited by quality flaws, which may lead to bias. The current study is an ecological study, and thus, the ecological bias is inevitable. Another economic factors or variables such as inflation, indicators of sustainable development etc. should be included in the further study.

Round 2
Reviewer 3 Report
After authors elaboration I agree. Only incorporate provided suggestions.